# A Neutropenic Diet in Haemato-Oncological Patients Receiving High-Dose Therapy and Hematopoietic Stem Cell Transplantation: A Systematic Review

**DOI:** 10.3390/nu17050768

**Published:** 2025-02-21

**Authors:** Luise Jahns, Jutta Hübner, Christina Mensger, Viktoria Mathies

**Affiliations:** 1Institute of Agricultural and Nutritional Sciences, Martin Luther University, 06120 Halle, Germany; 2Department of Hematology and Internal Oncology, University Hospital Jena, 07747 Jena, Germany; jutta.huebner@med.uni-jena.de (J.H.); christina.mensger@med.uni-jena.de (C.M.); viktoria.mathies@med.uni-jena.de (V.M.)

**Keywords:** neutropenic diet, low-bacterial diet, cancer, haematological malignancies, high-dose chemotherapy, stem cell transplantation, neutropenia, nutrition, nutrition therapy, systematic review

## Abstract

Background/Objectives: Although the benefits of low-germ diets for patients are increasingly being questioned, their application in practice is widespread. The aim of this review is to summarise the current data and evaluate the effectiveness of the neutropenic diet (ND) in adult haemato-oncological patients to provide a basis for practical guidelines. Methods: A systematic search was conducted in four electronic databases (Medline (Ovid), CINAHL (EBSCO), EMBASE (Ovid) and Cochrane CENTRAL) to identify English and German randomised controlled trials (RCTs) concerning the effectiveness of an ND in adult haematological patients. The main endpoints were fever and systemic infections, gastrointestinal (GI) infections, mortality, nutritional status and hospitalisation length. Results: A total of five RCTs with 510 adult patients were included in this systematic review. All patients received high-dose chemotherapy in order to treat haemato-oncological malignancies. None of the analysed endpoints showed a significant advantage of the ND compared to the control group. Conclusions: An ND does not have a beneficial effect on infection rates, GI health, mortality or hospitalisation length for haemato-oncological patients. On the contrary, an ND tends to negatively affect the patient’s nutritional status; therefore, an adaption in clinical routine should take place.

## 1. Introduction

Haematological malignancies represent a substantial global disease burden, with over one million new cases reported in 2020, according to the Global Cancer Observatory [1]. Treatment approaches vary and are tailored to the type of cancer, its stage and individual patient factors [2]. For aggressive forms of these diseases, including leukaemia, lymphoma, and myeloproliferative and myelodysplastic disorders, high-dose chemotherapy and haematopoietic stem cell transplantation (HSCT) remain the standard care [3].

Intensive chemotherapy, prior to the transplantation, serves as a conditioning regimen and has the purpose of inducing immunosuppression, which is required to enable engraftment [4,5]. Additionally, conditioning aims to eliminate malignant cells and induce myeloablation, therefore providing stem cell niches [5,6]. During this phase, patients experience long-term neutropenia, and numerous precautionary measures must be taken to minimise the risk of infection [7]. These include a protective environment, antimicrobial prophylaxis, intensive personal hygiene, strict food hygiene and the implementation of low-germ diets. Despite these precautions, infections still represent a major cause of morbidity and mortality [7,8].

A survey carried out by the European Society for Blood and Marrow Transplantation in 2015 and 2016 shows that over 90% of the centres in 23 countries under review use a “neutropenic diet” (ND) or a “low-bacterial diet” (LBD) as standard care in the early post-HSCT period [9]. In Germany, an LBD is recommended at 17 out of 26 university hospitals [10]. In a survey published in 2014, nutritionists in the UK were questioned about prescribing dietary restrictions to patients with chemotherapy-induced neutropenia [11]. The majority (67.8%) of the dietitians stated to recommend an ND and less than half (43.6%) had a policy in place for the implementation [11]. Similar to these findings, a survey performed in China showed that 83% of the centres considered an ND as part of the management for HSCT patients, but only 1 out of 18 institutions reported having a formal protocol [12].

The underlying idea of the ND originates from the 1960s, when a germ-free diet was introduced, focusing on sterilising all foods [13]. The concept is based on the fact that certain types of food, especially raw and unprocessed products like fresh fruit and vegetables or raw meat, are heavily contaminated with potentially pathogenic germs [13,14]. It has been postulated that the omission of these foods contributes to the prevention of infections; thus, neutropenic patients should avoid these or ensure sufficient heating to kill these organisms. There is no universal definition of the diet and the associated restrictions as well as time of implementation differ [8,14]. Typically excluded products are fresh fruit and vegetables, some types of cheese, uncooked meat, fish and eggs, as well as unpasteurized dairy products, but the limitations are often more extensive [14]. Hospitals and clinics hand out long lists of forbidden foods and dishes to their patients, which results in substantial dietary restrictions [10].

The restrictions on food choices and the recommendation to consume almost exclusively cooked foods leads to less nutritious and visually less appealing meals [15]. As a result, adequate enteral nutrition is very challenging or even impossible for some patients. Because of the debilitating disease and therapy, which is associated with complications like loss of appetite, nausea and mucositis, these patients are already at high risk for malnourishment [7,16,17]. Studies show that the nutritional status of recipients of autologous or allogeneic HSCT deteriorates particularly during hospitalisation, thus underlining the importance of nutritional care during this period [18,19].

Although there are major differences in the use of the ND as nutritional therapy, this concept is still considered standard care in many centres and institutions [9,10,20,21]. While widely used, there is a lack of evidence regarding the effectiveness of this diet. In 2009, the American Society for Blood and Marrow Transplantation updated the guidelines for preventing infectious complications among HSCT recipients [22]. While an LBD is recommended, it is pointed out that its efficacy in preventing infections is not adequately proven by studies [22]. The same year, the American Society for Parenteral and Enteral Nutrition published a clinical guide, also recommending dietary restrictions and the avoidance of high-risk foods while underlining the heterogenous data regarding the effectiveness [16]. More recent trials indicate that the ND is not useful to prevent infections and, on the contrary, may have negative impacts on the patient’s health and recovery [7,14,17].

These progressions were taken into account when updating the guidelines, as the European Society for Clinical Nutrition and Metabolism stated in 2021: “there was no evidence to support the use of LBD” [23]. In the last two years, two new randomised controlled trials (RCTs) [24,25] have been published investigating this issue. To our knowledge, these studies have not yet been included in any other systematic review [7,8,14,17,26,27]. The aim of this systematic review is to summarise the current data and evaluate whether an ND provides an advantage or disadvantage in the treatment of haemato-oncological and HSCT patients. The primary objective of the study was to study the incidence of infection while the secondary objectives were to investigate GI health, mortality, hospitalisation length and nutritional status. A further aim is to provide a scientific basis for treatment and initiate adaptations in clinical practice.

## 2. Materials and Methods

This review follows the PRISMA guidelines [28] and is registered with OSF (registration DOI: https://doi.org/10.17605/OSF.IO/ZUW3E, accessed on 1 February 2025. The PRISMA checklists are provided in the Appendix A.

### 2.1. Criteria for Including and Excluding Studies in the Review

The inclusion and exclusion criteria are listed in Table 1 based on the PICO model [29]. Only RCTs were included, as they provide results at the highest level of evidence. All interventions based on the ND compared with an unrestricted diet were considered. The selection of studies was limited to publications that reported patient-relevant outcomes such as infections, mortality and nutritional status in haemato-oncological patients. Additionally, since children and adults are clinically too heterogeneous, paediatric trials were excluded. Language restrictions were made to English and German.

### 2.2. Study Selection

A systematic search was conducted using four databases (Medline (Ovid), CINAHL (EBSCO), EMBASE (Ovid) and Cochrane CENTRAL) on 17 April 2024. For each database a complex search strategy was developed consisting of a combination of MeSH terms, keywords and text words with different spellings related to haematological cancer and neutropenic diets. The detailed search string is provided in the Appendix A. No references from additional sources were used. After importing the search results into EndNote20, all duplicates were removed, and a title–-abstract screening was carried out independently by two reviewers (VM and LJ). In case of disagreement, consensus was reached by discussion; subsequently, all full texts were analysed and their eligibility with regard to the research question was determined. The study flow during this process is presented in Figure 1, according to the PRISMA reporting statement [30].

### 2.3. Data Extraction

Data extraction was performed by one reviewer (LJ) and controlled independently by three reviewers (VM, CM and JH). We extracted data about the number of participants, cancer type, type and duration of intervention, and all endpoints and outcomes from each study (Table 2).

### 2.4. Assessment of Risk of Bias and Methodological Quality

The risk of bias in the included studies was analysed with the Cochrane revised Risk of Bias Tool 2.0 [31]. All characteristics were assessed independently by two reviewers (VM and LJ). In case of disagreement, a third reviewer was consulted (JH) and consensus was reached by discussion.

## 3. Results

### 3.1. Characteristics of Included Studies

The systematic research revealed 3970 results. No studies were added by hand search. First, duplicates were removed, leaving 2723 studies. After screening the titles and abstracts, 19 studies remained to complete the review (Figure 1). Finally, five publications were analysed in this review, all of which are RCTs. Detailed characterisation of the included studies can be found in Table 2.

Radhakrishnan et al. included 131 paediatric patients, but a separate evaluation between paediatric and adult patients was carried out by the study authors; therefore, only adults are considered in this review [27].

In total, 510 adult patients were included in the five studies of this review. One study did not provide any information on the gender distribution of the participants [33]; the remaining RCTs included slightly more men (199 men vs. 158 women). Patient ages ranged from 17 to 88 years.

Permitted and prohibited foods varied considerably between studies. Raw fruit and vegetables were not allowed in all studies, with the exception of thick-skinned fruit in two publications [7,32]. Raw animal products such as meat, fish, eggs and unpasteurized dairy products were excluded in three trials [7,27,34]. The diets of the control groups were also heterogenous, ranging from encouragement to eat fresh fruits daily [27,33] and normal hospital diets [34] to moderate limitations [27]. In four of the five studies, the control diet included basic hygiene practices for safe food handling [7,27,32,33].

### 3.2. Excluded Studies

One RCT was excluded due to the inclusion of paediatric patients, where a separate analysis was not conducted. No full text was available for one publication. Twelve publications were excluded due to the publication type; seven of these are SRs (however, they were checked for additional RCTs), four are narrative reviews and one is a descriptive pilot study. Excluded publications with detailed exclusion reasons can be found in the Appendix A.

### 3.3. Risk of Bias in Included Studies

Methodical quality was assessed using the Risk of Bias 2.0 Tool for RCTs. All the included studies show medium to high risk (Appendix A, Figure 2).

### 3.4. Major Endpoints

#### 3.4.1. Fever and (Systemic) Infections

Infections, which are the main reason to implement an ND, were analysed in all five studies. However, the studies defined infections differently. Due to the compromised immune system of the patients, fever remains one of the few visible symptoms of an infection and, in some cases, the terms are used synonymously. As the origin of the infection or fever often cannot be determined, it is referred to as fever unknown origin (FUO) [30].

Van Tiel et al. defined infections as a temperature ≥ 38.5 °C or <36.0 °C [31]. The median number of days per chemotherapy cycle with temperature ≥ 38.5 °C or <36.0 °C was three days in the LBD arm vs. six days in the normal diet arm (*p* = 0.11). The number of chemotherapy cycles with infection in relation to the total number of cycles did not differ significantly between groups (*p* = 0.48). In each group, one patient was diagnosed with candidemia and two cases of possible invasive aspergillosis occurred in the non-restrictive group (n = 9) [31].

Stella et al. differentiated between infection of grade ≥ 2, incidence of sepsis and FUO [25]. In none of these categories was the ND significantly superior to the control group. Infections of grade ≥ 2 occurred in 65% of the ND arm vs. 61% of the control arm (*p* = 0.7). Sepsis occurred in 9% of the patients adhering to the ND vs. 13% in the non-restrictive group (*p* = 0.4). The fever rates were 43% in the ND arm and 33% in the non-restrictive arm (*p* = 0.1) [25].

Radhakrishnan et al. included patients between 1 and 60 years of age [24]. Only the results for adult patients are presented below. Blood cultures, tissue and fluid cultures and imaging studies were used for evaluation as well as specialised tests for detecting viral infections. No significant difference was found regarding major infections comparing regular diets and the ND (32% vs. 25%, *p* = 0.26) [24].

Lassiter et al. determined the incidence of infections only by blood cultures and found no significant difference between the ND and the unrestricted diet (28% vs. 30%, *p* = 0.99) [33].

The study from Gardner et al. showed a non-significant difference, with 51% of the patients developing FUO in the cooked diet and 36% in the control group (*p* = 0.07) [32]. In terms of infections, a subdivision was made into major infections (pneumonia, bacteraemia or fungemia, or pneumonia accompanied by bacteraemia or fungemia) and minor infections. So significant difference was found in either subgroup, comparing the LBD with the control group (29% vs. 35%, *p* = 0.6; 6% vs. 5%, *p* = 0.99) [32].

#### 3.4.2. Gastrointestinal Infections/Complications

Many complications of therapy are connected with conditions affecting the GI tract [34]. Since they represent a major factor for the course of the disease, the influence of nutrition on infections and intestinal microbiota is of great importance [34].

Stella et al. found no significant differences regarding gut infection when comparing the LBD and a regular diet [25]. In both arms, 14% of the patients developed a GI infection (*p* > 0.99). Mucositis was diagnosed in 67% of the patients adhering to the LBD and in 71% following no dietary restrictions (*p* = 0.6) [25].

The influence of the ND on gut colonisation by yeasts and aerobic Gram-negative bacilli was examined by van Tiel et al. [31]. An LBD could not significantly lower the value of yeast CFU (cycle 1: *p* = 0.6; cycle 2: *p* = 0.14) or aerobic Gram-negative bacilli CFU (cycle 1: *p* = 0.43; cycle 2: *p* = 0.26) over the period under observation. Furthermore, Radhakrishnan et al. observed no significant increase in stool culture positivity comparing an ND and a regular diet (*p* = 0.13) [24]. On day 15 of induction, 47% of those following a regular diet and 28% of those in the ND group tested positive [30].

#### 3.4.3. Mortality

The mortality rates were assessed by three studies. Stella et al. reported one death during the period of neutropenia (at day 30) in the non-restrictive arm, which was secondary to multiorgan failure, developed in the context of a cytokine release syndrome [25]. Gardner et al. compared the influence of the ND compared to a non-restrictive diet on death over a period of three years and found no significant difference in the probability of survival between the two groups (*p* = 0.36) [32].

Radhakrishnan et al. analysed the induction mortality rates and detected no significant difference (18% vs. 11%, *p* = 0.46) [24]. While following a regular diet, six patients died, five of them due to sepsis and one due to progressive disease. In the ND arm, three patients died due to sepsis and one due to progressive disease [24].

#### 3.4.4. Nutritional Parameters

Nutrition has considerable influence on the course of the disease and a patient’s quality of life. Therefore, it is a major factor when assessing the usefulness of the ND.

Lassiter et al. used the Patient-Generated Subjective Global Assessment Tool (PG-SGA). The exclusively graphical depiction shows a similarly strong increase in the score in both groups, with a higher score in the ND group in tendency. The authors report that 60% in either group required total parenteral support [33].

Regarding nutritional status, Stella et al. examined body weight loss, serum albumin variations and the use of enteral/parenteral nutrition in their patient population [25]. While the non-restrictive group had a significantly lower mean percentage of weight loss after one month, compared to the LBD (*p* = 0.03), no differences between groups were found in body weight variation (kg) from admittance to discharge (*p* = 0.3). In addition, changes in Body Mass Index from admittance to discharge did not significantly differ between the two study arms (*p* = 0.7). The ND had no significant effect on the serum albumin variation from admission to discharge. In the ND arm, the serum albumin level fell by 14.5% and by 18% in the control arm (*p* = 0.1). In terms of nutritional support, parenteral nutrition was preferred over enteral nutrition. In the ND group, 23% of the intervention group and 26% of the control group received parenteral nutrition (*p* = 0.8). Furthermore, the duration of parenteral support did not differ significantly between arms (*p* = 0.8) [25]. For more information on nutritional behaviour, patients kept a diary on clinical symptoms, since they can limit adherence. The symptoms nausea and diarrhoea occurred equally in both groups. Under the regular diet, 15% reported nausea, and this was 17% in the ND arm (*p* = 0.9). Diarrhoea was reported by 34% of the control group and 31% of the ND group (*p* = 0.7) [25].

##### Nutrition-Related QoL

Stella et al. evaluated the effect of each diet on QoL by assessing the satisfaction of patients based on the diaries [25]. In total, 16% of patients following an ND and 35% following an RD stated that the prescribed diet did not negatively affect alimentation (*p* = 0.003) [25].

##### Hospitalisation Length

The duration of hospital stay was analysed by one study. Stella et al. reported no significant difference between the two arms. Patients adhering to the regular diet had a mean hospitalisation length of 22 days and patients following the ND had a mean hospitalisation length of 21 days (*p* = 0.6) [25].

## 4. Discussion

### 4.1. Diets

A crucial point in the debate regarding LBD is the lack of standardisation. As there is no official definition, each institution implements the diet differently, both in terms of food selection and duration. This also poses a problem for the comparability of studies, as there are major differences, including in the RCTs considered in this review. While three studies did not allow any fresh fruit and vegetables [24,31,32], thick peeled fruits were permitted in two studies [25,33]. There were further differences regarding nuts, honey, spices and tap water. Further, the control diets were inconsistent with regard to permitted foods and hygiene measures.

### 4.2. Fever and Systemic Infections

Fever and infection rates were assessed by all five included RCTs. All studies showed no significant difference between an ND and the control group in these endpoints [24,25,31,32,33].

Despite the homogeneous results, comparability is limited due to the heterogeneous assessment methods.

Van Tiel et al. defined infection only by body temperature (≥38.5 °C or ≤36.0 °C) with a single measurement [31]. While high body temperature can indicate an infection, it is not sufficient on its own for a diagnosis. The absence of fever does not preclude an infection since the immune system (especially in HSCT patients) is often compromised and therefore unable to develop fever [30]. Furthermore, only 20 patients were included; therefore, the significance of the study is limited.

Stella et al. differentiated between infection, sepsis and FUO, but no detailed information was given regarding the definition and assessment of these categories [25].

Despite the methodological shortcoming, these findings are consistent with other research. Jakob et al. conducted a case–control study in haemato-oncological patients to assess the association between a standard diet compared to an ND and infection-related endpoints [35]. The standard diet did not affect days of fever or positive blood culture. Also, Heng et al. found no association of an ND and the prevention of infection-related endpoints [36]. Another retrospective study showed even higher infection rates in patients who followed an ND [37]. Taken together, these results strongly indicate that an ND is not superior to a control diet in terms of infection prevention.

### 4.3. Gastrointestinal Infections

The basic idea behind the low-germ diet is based on the assumption that a lower microbial food load leads to fewer infections, since the mucosa is damaged through chemotherapy and microbial translocation is promoted. At the time the diet was introduced, the concept seemed valid, as fresh fruit and vegetables had been identified as sources of opportunistic pathogens [38] and animal studies reported better outcomes in germ-free mice [39]. However, the research does not support the theory. The publications analysed in this review showed no evidence that an ND can prevent GI infections [24,25,31]. Furthermore, in the retrospective study by Trifilio et al., GI infections were more common in the ND group [37].

Based on the findings of this review and additional research, the concept needs to be reconsidered, especially as there is a better understanding of host–microbe interaction. Research in the field of the intestinal microbiome is progressing and the concept of “colonisation resistance” is increasingly mentioned [40]. The preservation of microbiome diversity is essential to maintain protective functions and inhibit pathogenicity [37]. Therefore, a reduced intake of microorganisms and dietary fibre in the context of an ND would even increase the risk for infection [37,41,42].

### 4.4. Mortality

None of the RCTs included in this review found a significant effect of the ND on mortality. It should be noted that two of the studies only covered a short period of time [24,25], while Gardner et al. recorded mortality rates over three years [32]. Although a short observation period limits the informative value, all these results are in line with other research. In a retrospective study of HSCT patients, no difference in mortality rates between an ND and a control diet was found [37]. Another retrospective study in haemato-oncological patients also found no association between an ND and death within 28 days [35].

### 4.5. Nutrition

The influence of the ND on the nutritional status was assessed in only two studies. Lassiter et al. used the Patient-Generated Subjective Global Assessment and reported an equal increase in the score in both groups, where high scores indicate a higher risk for malnutrition [33]. Stella et al. assessed variations in body weight loss, serum albumin, use of enteral/parenteral nutrition, nausea and diarrhoea. None of these parameters were affected by implementing an ND, but patients described a negative effect on their alimentation significantly more often when following the ND [25].

However, a case–control study indicates nutrition-related problems more clearly. Jakob et al. reported significantly fewer cases of diarrhoea and a trend towards less weight loss in patients following a standard diet compared to the ND [35]. In a retrospective study comparing a normal hospital diet with an ND, diarrhoea was also more common in the ND group (*p* < 0.095) [37].

An ND tends to have a lower content of vitamins and minerals due to the exclusion of fresh fruits, vegetables and cooking-induced nutrient losses [15,43]. One possible explanation for the fact that this could not be shown in the RCTs included in this review concerns the period under consideration. In both studies, the intervention was carried out for the period of neutropenia or until discharge [25,33]. Stella et al. reported a median of six days [25]. Such a short period of time is not sufficient to represent corresponding changes in nutritional status. In addition, insufficient food intake was compensated by parenteral nutrition in both RCTs [25,33].

Food choice and nutritional content are often restricted by an ND and adequate food intake is made more difficult for patients. Since an ND hinders adequate nutrition, parenteral nutrition may become necessary to ensure sufficient nutritional intake, even though the guidelines state that enteral nutrition should be preferred over parenteral nutrition due to the higher risk of side effects [16,44]. Therefore, a regular diet according to safe food handling practices appears to be beneficial and is more compatible with the guidelines.

### 4.6. Quality of Life

One RCT analysed the impact of an ND on hospitalisation length and found no significant difference [25]. Only one further study could be identified, which assessed the effect in a comparable setting. The retrospective study by Trifilio et al. also detected a similar duration of hospitalisation comparing an ND and a normal hospital diet [37].

In addition to the function of nutrient supply, nutrition also has a considerable influence on a patient’s QoL. Patients experience symptoms such as loss of appetite and changes in taste. The restriction of food choices, therefore, exacerbates these challenges. Especially odourless and cool foods, which are appealing choices for patients suffering from chemotherapy side effects, are often excluded by an ND. It is also very demanding to follow such a strict diet and can cause additional psychological stress. In accordance with this, Stella et al. reported higher satisfaction in the non-restrictive arm [25].

### 4.7. Strengths

The strength of this systematic review is that, in contrast to the existing SRs on this topic [7,8,14,17,26,27,45], two new RCTs were analysed [24,25]. The exclusive analysis of RCTs with adult patients provides a qualitative advantage compared to other publications (Appendix A). Additionally, we assessed the risk of bias of the included RCTs using the Risk of Bias 2.0 Tool. The analysis pointed out methodological weak points of the included trials, which should be considered when assessing the significance of this review.

### 4.8. Limitations

This systematic review has limitations that must be mentioned. Only studies in English or German language were included and, overall, only a small number of studies could be included in this review. Additionally, conducting a meta-analysis was not feasible due to the heterogeneity of the included studies.

### 4.9. Implications for Clinical Practice

In clinical practice, the focus should be directed towards individualised nutritional care. Taking into account individual preferences or aversions and possible side effects of therapy, attention should be paid to providing patients with the best possible nutritional care.

In order to ensure the prevention of infection, the education of patients, relatives and medical staff regarding safe food handling is of central importance. In addition, the scientific education of medical personnel and nutritionists regarding the current data availability should be promoted. The updated guidelines on nutrition for immunocompromised patients should be communicated more effectively and consistently implemented. Hence, outdated therapy strategies can be avoided and standardised, and scientifically based nutritional practices can be established.

## 5. Conclusions

To summarise, the evidence does not support the conclusion that the implementation of an ND reduces the risk of systemic and GI infections, mortality or length of hospitalisation. Furthermore, the reviewed publications indicate an increased risk of malnutrition, as patients following an ND more frequently describe a negative influence on nutritional behaviour. Therefore, an ND is obsolete and causes unnecessary restrictions. Safe food handling practices prove to be a sufficient alternative.

Clinics and caregivers that continue to apply the ND are acting contrary to the current state of research and guidelines and endanger the recovery and QoL of patients. Education and adaptation in clinical practice must be promoted to avoid incorrect nutritional supply to this vulnerable group.

## Figures and Tables

**Figure 1 nutrients-17-00768-f001:**
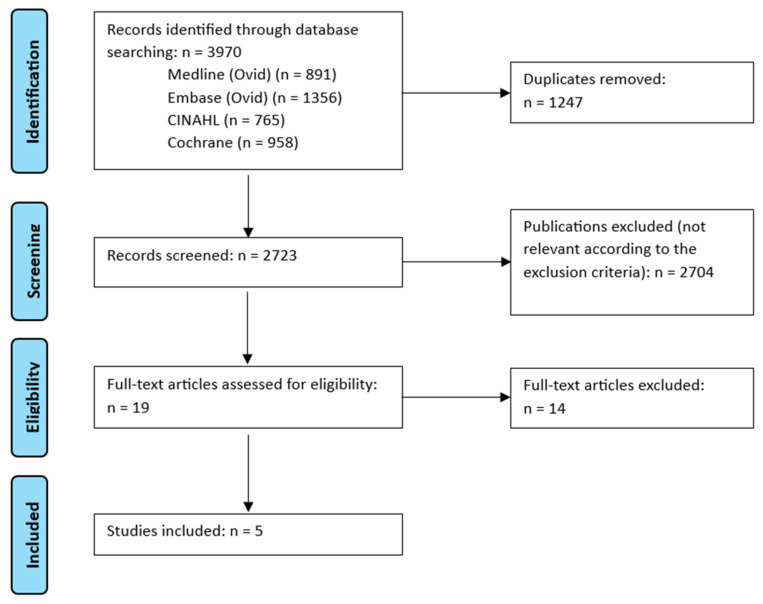
Flowchart summarising the study selection process according to the PRISMA reporting statement [28].

**Figure 2 nutrients-17-00768-f002:**
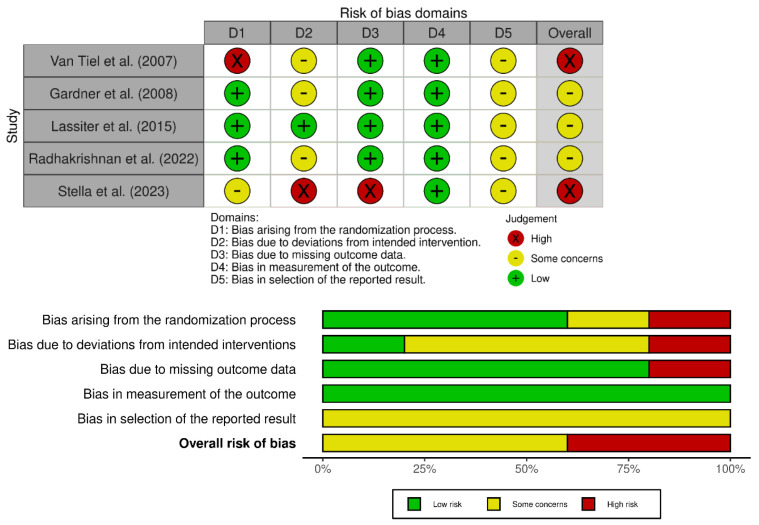
Risk of Bias 2.0 summary (generated with Robvis tool) [24,25,31,32,33].

**Table 1 nutrients-17-00768-t001:** Inclusion and exclusion criteria based on the PICO model.

PICO	Inclusion Criteria	Exclusion Criteria
Patient	Adult haemato-oncological patients	Studies exclusively with paediatric patients
Intervention	Every intervention based on neutropenic diet No restriction regarding the type of neutropenic diet	-
Comparison	All unrestricted diets, less restrictive diets, diets according to safe food handling recommendations, normal hospital diet	-
Outcome	Infections (fever, GI infections), mortality, nutritional status, QoL	Outcomes not relevant to patients
Study	Randomised controlled trials (RCTs)	Systematic reviews; umbrella reviews
Others	Language: German and English Full publication	Grey literature (conference articles, abstracts, letters, ongoing studies, unpublished research, etc.) Full text not available in German or English

Abbreviations: GI—gastrointestinal; QoL—quality of life.

**Table 2 nutrients-17-00768-t002:** Study characteristics.

Reference	n/Cancer Type	Intervention	Duration	Endpoints	Outcomes
van Tiel et al. (2007) [31]	n = 20 AML, ALL	**Normal hospital diet** (n = 10): no detailed information **Low-bacterial diet** (n = 10): not allowed are raw vegetables, salads, soft cheeses, raw meat products, most fresh fruits, tap water and spices added after cooking; bread, cheese and ham are individually packed; yoghurt desserts, soda drinks and soups are served in single-serving containers **All patients**: antimicrobial prophylaxis	no information	**1**: colonisation of the digestive tract with aerobic Gram-negative bacilli and yeasts **2**: infections (number of chemotherapy cycles with infection median number of days with temp. ≥ 38.5 °C or <36.0 °C	**Normal hospital diet vs. low-bacterial diet** **1**: not significantly different between treatment groups; first chemotherapy cycle: *p* = 0.42, second cycle: *p* = 0.26 **2**: no significant difference: chemotherapy cycles with infections/total number of treatment cycles: 17/21 vs. 14/20 (*p* = 0.48) **3**: no significant difference: 6 vs. 3 (*p* = 0.11)
Gardner et al. (2008) [32]	n = 153 AML/MDS	**Cooked diet** (n = 78): no raw fruits and no raw vegetables **Raw diet** (n = 75): fresh (raw) fruits and vegetables permitted (washed with cold water for 30 s); patients were encouraged to eat fresh fruit and vegetables at least once daily **All patients**: protected environment (air-filtered rooms); antibacterial and antifungal prophylaxis	until discharge or max. six weeks	**1**: major infection **2**: survival of three years **3**: minor infections **4**: FUO	**Cooked diet vs. raw diet** **1**: no significant difference between groups: 29% vs. 35% (*p* = 0.6) **2**: no significant difference between groups: 46/78 (59%) vs. 41/75 (55%), (*p* = 0.36) **3**: no significant differences: 6% vs. 5% (*p* = 0.99) **4**: no significant differences: 51% vs. 36% (*p* = 0.07)
Lassiter et al. (2015) [33]	n = 46 various types of leukaemia, lymphoma and myeloma	**ND** (n = 25): only cooked food and thick-skinned fruits **Unrestricted diet** (n = 21): no restrictions **All patients**: instructed to follow safe food handling (according to FDA); protected environment (air-filtered room); antibiotic prophylaxis	until end of neutropenia or until discharge	**1**: incidence of infection **2**: nutritional status (PG-SGA tool)	**ND vs. unrestrictive diet** **1**: no difference in percentage of positive blood cultures: 7/25 (28%) vs. 6/20 (30%), (*p* = 0.99) **2**: according to authors: does not appear to be significant difference (presentation of PG-SGA scores as graph only)
Radhakrishnan et al. (2022) [24] (only adults analysed)	n = 69 AML, ALL	**ND** (n = 35): no raw fruit, no raw juices, no raw vegetables **Regular diet** (n = 34): patients were encouraged to consume a min. of one serving of raw fruit or vegetable per day **All patients**: uncooked fish, uncooked meat, uncooked eggs and raw nuts were not allowed; pasteurised dairy products (milk, yoghurt) were allowed; antifungal prophylaxis; food safety guidelines were followed	until completion of induction and discharge or day 40	**1**: major infections **2**: stool microbial flora **3**: induction mortality rates	**Regular diet vs. ND** **1**: no significant difference between the groups, 41% vs. 37% (*p* = 0.73) **2**: no impact on positive stool cultures on day 15: 47% vs. 28% (*p* = 0.13) **3**: No significant difference between the groups, 18% vs. 11% (*p* = 0.46)
Stella et al. (2023) [25]	n = 222 lymphomas, multiple myelomas, AML, other	**Protective diet** (n = 111): allowed are cooked fish, meat and vegetables; washed and peeled thick peeled fruit; pasteurised milk and cheese; freeze-dried eggs; no yoghurt, honey, cold cuts and sausages **Non-restrictive diet** (n = 111): allowed are cooked fish, meat and eggs; fresh fruit and vegetables (manipulated according to safe food handling procedures); pasteurised milk, honey, yoghurt and cheese (without mould) **All patients:** bread is allowed; only industrial prepared desserts	from start of chemotherapy, during period of severe neutropenia (ANC < 550/µL)	**1**: incidence of infection of grade ≥ 2 **2**: death during period of neutropenia **3**: incidence of GI-tract infections **4**: incidence of FUO **5**: nutritional status **6**: use and duration of PN **7**: duration of hospital stay [d] **8**: QoL (negative effect on alimentation) **9**: estimated overall survival at 30 days from the onset of neutropenia **10**: acute GVHD (in allo-HSCT patients)	**Protective diet vs. non-restrictive diet** **1**: no significant difference, 65% vs. 61% (*p* = 0.7) **2**: one death in non-restrictive arm secondary to multiorgan failure **3**: no significant difference, 14% vs. 14% (*p* > 0.99) **4**: no significant difference, 43% vs. 33% (*p* = 0.1) **5**: body weight loss (1 month): −4.6% vs. −3.4% (*p* = 0.03); −3.7 kg vs. −2.7 kg (*p* = 0.04); no differences between the arms from admittance to discharge (−3.6 kg vs. −3.2 kg, *p* = 0.3); albumin variation from admittance to discharge: −14.5% vs. −18% (*p* = 0.1); BMI variation from admittance to discharge: −0.9 vs. −0.8 (*p* = 0.7) **6**: use of PN: 23% vs. 26% (*p* = 0.8); duration of PN: 6.9 days vs. 6.7 days (*p* = 0.8) **7**: no significant difference: 21 vs. 22 (*p* = 0.6), presentation of patient-reported QoL as a graph only **8:** 16% vs. 35% (*p* = 0.003) **9**: no data **10**: aGVHD (any grade): 35% vs. 29% (*p* = 0.4)

Abbreviations: AML—acute myeloid leukaemia; ALL—acute lymphoblastic leukaemia; MDS—myelodysplastic syndrome; FUO—fever unknown origin; PG-SGA—Patient-Generated Subjective Global Assessment; ANC—absolute neutrophil count; GI—gastrointestinal; PN—parenteral nutrition; QoL—quality of life; aGVHD—acute graft-versus-host disease; HSCT—hematopoietic stem cell transplantation; BMI—body mass index

## Data Availability

The data presented in this study are available on request from the corresponding author. The data are not publicly available due to technical limitations.

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
