# Peer review of "A Neutropenic Diet in Haemato-Oncological Patients Receiving High-Dose Therapy and Hematopoietic Stem Cell Transplantation: A Systematic Review"

_nutrients, 2025, doi:10.3390/nu17050768_

Round 1
Reviewer 1 Report
Comments and Suggestions for Authors
Dear Authors,
I believe that this manuscript addresses an important topic and aligns well with the journal’s objectives. Below are my comments and suggestions:
INTRODUCTION
- The introduction is well-structured and clear; however, I suggest incorporating a section discussing the nutritional status of this population, as they are often at high risk of malnutrition.
- Consider adding a paragraph exploring the evolution of nutritional status in patients undergoing autologous and allogeneic hematopoietic cell transplantation, which could provide a relevant basis before moving on to the specific objective of your study.
METHODS
- I recommend specifying at the beginning of this section that the study has been reported following PRISMA guidelines.
- Authors should provide the PRISMA checklist as a supplementary file.
- As this is a systematic review, the protocol should be registered in PROSPERO or OSF, as this is a crucial aspect for transparency and methodological rigor.
RESULTS, DISCUSSION & CONCLUSION
- These sections are well-presented, and I commend the authors for their clarity and depth.
- Following the limitations section, I recommend adding a paragraph on "Implications for Clinical Practice", as this would strengthen the study’s real-world applicability.
I believe that after these modifications, the study can be considered for potential publication.
Reviewer 2 Report
Comments and Suggestions for Authors
The authors of this manuscript (systematic review) analyzed two more studies-randomised controlled trials published in recent years, which could (according to the researchers) supplement our knowledge regarding recommendations for the use of a neutropenic diet in patients after megachemotherapy. The authors analyzed, based on previous and new observations, the pros and cons of using a neutropenic diet in relation to the occurrence of infections/fever, gastrointestinal problems, nutritional status, quality of life, mortality. In general, the manuscript is well prepared, but…Taking into consideration the different results of the analyzed reports and the lack of clear, convincing results for or against the neutropenic diet, I believe that this analysis adds to little and does not clarify ourcurrent knowledge regarding the importance of a neutropenic diet.
Reviewer 3 Report
Comments and Suggestions for Authors
Dear Authors, thank you for your work. I read with interest your paper. I'm looking for Prospero registration, but I can't find it. what about other inclusion/exclusion criteria (presence of full text, language of publication, publication year, etc)?
Reviewer 4 Report
Comments and Suggestions for Authors
Dear Authors,
the comments in the annex file.
Best.

Round 2
Reviewer 2 Report
Comments and Suggestions for Authors I accept the content of the manuscriptAuthor Response
Comment: I accept the content of the manuscript.
Reply: Dear reviewer, thank you very much for taking the time again to review and accepting our manuscript.
Reviewer 4 Report
Comments and Suggestions for Authors
Dear Authors,
the comments in the annex file.
Best

Native Review required
Author Response
Comment:
Introduction: I believe it can be improved by significantly reducing the text (at least by 50%) and adding an
epidemiological approach to the phenomenon at the beginning of the section, both globally and locally.
Response:
Dear reviewer,
thank you very much for taking the time again to review our manuscript. Based on your suggestions, we have revised the introduction by shortening key sections, including the medical background. We also restructured it to present epidemiological information on hematological malignancies and the neutropenic diet at the beginning. We hope these changes align with your suggestions and enhance the manuscript. Additionally, we have carefully reviewed the language and references to ensure clarity and accuracy.